# The Procedure of Identifying the Geometrical Layout of an Exploited Railway Route Based on the Determined Curvature of the Track Axis

**DOI:** 10.3390/s23010274

**Published:** 2022-12-27

**Authors:** Wladyslaw Koc

**Affiliations:** Faculty of Civil and Environmental Engineering, Gdansk University of Technology, 80-233 Gdansk, Poland; kocwl@pg.edu.pl

**Keywords:** railway track, curvature of the track axis, moving chord method, identification of geometric layout

## Abstract

This paper presents a detailed algorithm for determining the curvature of a track axis with the use of a moving chord method, and then discusses the procedure for identifying the geometric layout of an exploited railway route on the basis of the determined curvature. In the moving chord method, the measured coordinates of the track axis allow one to directly determine the existence of the horizontal curvature without the need for additional measurements. This enables comprehensively identifying the existing geometric elements—straight lines, circular arcs, and transition curves. The conducted activities were illustrated with a calculation example, including a 5.5 km long test section with five areas of directional change. This showed a sequential procedure that led to the solution of the given problem. Based on the curvature diagram, the coordinates of the segmentation points, which define the connections of the existing geometric elements with each other, were determined.

## 1. Introduction

Identification of the geometric layout of a railway route consists in determining the location of the existing straight and curved sections, as well as determining the appropriate numerical parameters (turning angles of the route, radii of circular arcs, and lengths of transition curves). Of course, the above data are included in the design documentation of the railway line, but operational factors and maintenance processes mean that the validity of the documentation may be limited and it becomes necessary to verify it periodically. To do this, you need the right tools.

Determining the geometric shape of a railway route is based on the measured coordinates of the track axis in the linear and Cartesian reference system. The currently used measurement methods are similar in different railway administrations [1,2,3,4,5,6,7,8]. In classic geodetic techniques, distances and angles are measured using tachymeters in relation to the spatial geodetic network. Further possibilities are provided by stationary satellite measurements based on the global navigation satellite system (GNSS) technique. This solution does not require using the point network of the railway geodetic network; the measurement systems use the so-called active geodetic networks (e.g., networks of reference stations operating in a real time network (RTK) [9,10,11]). Mobile satellite measurement methods are also being introduced, in which (apart from GNSS receivers) inertial navigation system (INS) devices [12] are used as supporting devices, as well as optical methods such as terrestrial laser scanning (TLS) [13]. Research is being conducted on the possibility of using systems consisting of satellite receivers mounted on various types of vehicles [14,15,16,17,18].

Determining the coordinates of the track axis makes it possible to visualize a given railway route, giving a general orientation of its location. However, since the purpose of the measurements is to determine the geometrical parameters (i.e., identification) of the measured route, appropriate calculation algorithms should be used (referring, for example, to the principles of the analytical design method [19,20,21,22]). As it turns out, the problem can be effectively solved by using the obtained measurement data to determine the existing curvature of the geometric layout. Papers [23,24,25] present relevant analysis relating to the proposed new method of determining the curvature of the track axis, referred to as the “moving chord method”. They concern the application of this method for model geometric layouts (described with mathematical equations). Papers [26,27,28] address the issue of its use for the estimation of the horizontal curvature of the axis of the exploited railway track based on Cartesian coordinates obtained by direct measurements.

This work presents a detailed procedure for identifying the chosen geometric layout of the exploited railway route based on the curvature of the track axis. The length of this route is 5.5 km, with five curved sections located on it. The computational algorithms presented in paper [27] were used to determine the curvature.

As a result of geodetic measurements, Cartesian coordinates of railway route points are determined in the appropriate national spatial reference system. In Poland, for plane coordinates, the PL-2000 system [29] is enforced, created based on a mathematically unequivocal assignment of points of the GRS 80 reference ellipsoid [30] to appropriate points on the plane according to the Gauss–Krüger mapping theory [31].

## 2. Determination of the Curvature of the Track Axis Using the Moving Chord Method

In the given case (i.e., examining the horizontal plane), the analysis is based on the determined values of the plane’s eastern coordinates *Y_i_* and the northern *X_i_* coordinates of a given measurement point in the PL-2000 system. However, the proper identification of the track axis is provided by the appropriate graphs referring to the length parameter *L*. Therefore, in order to create the possibility of further analysis, it is necessary to refer to the linear system, which means determining the distances (variable *L*) of individual measurement points from the chosen starting point *O*(*Y*_0_, *X*_0_) (i.e., point *i*_0_).

The distance between two consecutive measurement points is
(1)$$\Delta L_{i÷i+1}=\sqrt{{\left(Y_{i+1}-Y_{i}\right)}^{2}+{\left(X_{i+1}-X_{i}\right)}^{2}},\;\;i=1,2,\ldots ,n$$

The linear coordinate *L_i_*, i.e., the distance from the point *O*(*Y*_0_, *X*_0_), is determined from the formula
(2)$$L_{i}=\sum_{i=0}^{n-1}\Delta L_{i÷i+1}$$

From a practical point of view, it is beneficial to transfer the measurement data to the local *x*, *y* coordinate system. In most cases, this operation will consist of shifting the origin of the PL-2000 system to the chosen point *O*(*Y*_0_, *X*_0_). Then, the following transformation formulas are used:(3)$$x_{i}=Y_{i}-Y_{0}$$
(4)$$y_{i}=X_{i}-X_{0}$$

Additionally, sometimes it may be advisable to rotate this system by the angle *β*. The relevant formulas are then as follows:(5)$$x_{i}=\left(Y_{i}-Y_{0}\right)\cos\beta +\left(X_{i}-X_{0}\right)\sin\beta$$
(6)$$y_{i}=-\left(Y_{i}-Y_{0}\right)\sin\beta +\left(X_{i}-X_{0}\right)\cos\beta$$

The positive value of the angle *β* occurs when the system is rotated to the left.

In the local coordinate system, the linear coordinate *L_i_* determined by the Formula (2) still remains valid.

The methodology for determining the curvature of the track axis has been explained in detail in [27]. The sequence of actions to determine the curvature value at any measurement point is shown in Figure 1.

We start determining the curvature *κ_i_* from the measurement point *i*, which is located in such a way that it allows the projection of a virtual chord length *l_c_* backward; the end of calculations must take place at a point from which a virtual chord of the same length can still be placed forward. The basic operation that must be carried out first is determining the numbering of the points defining the intervals in which the ends of the virtual chords drawn from point *i* are located.

For a chord drawn from point *i* forward, the interval in which the end of the chord occurs is determined by the points *p_i_* − 1 and *p_i_* (Figure 1). We determine it by successively checking the distances between point *i* and consecutive measurement points, in accordance with the direction of increasing numbering. These distances are
(7)$$l_{i÷(i+k)}=\sqrt{{\left(x_{i}-x_{i+k}\right)}^{2}+{\left(y_{i}-y_{i+k}\right)}^{2}},\;\;k=1,2,\ldots$$

After each step of the calculations, we check whether the condition $l_{i÷(i+k)}\geq l_{c}$ has been met. The first value of *i* + *k* that meets the compulsory condition is marked *p_i_*. Since the coordinates of points *p_i_* − 1 and *p_i_* are known, it is possible to write the equation of a straight line that goes through these points analytically. This equation has the following form:(8)$$y=a_{pi}+b_{pi}x$$

As can be seen in Figure 1, the end of the front chord (i.e., point *P_i_*) lies on the straight line described by Equation (8), at a distance *l_c_* from point *i*. It is therefore the point of intersection of the circle with radius *l_c_* and center at point *i* with the straight line (8). The coordinates of the *P_i_* point are determined from the following formulas:(9)$$x_{Pi}=\frac{-B_{Pi}\pm \sqrt{B_{Pi}^{2}-4A_{Pi}C_{Pi}}}{2A_{Pi}}$$
(10)$$y_{Pi}=a_{pi}+b_{pi}\frac{-B_{Pi}\pm \sqrt{B_{Pi}^{2}-4A_{Pi}C_{Pi}}}{2A_{Pi}}$$
where
$$\begin{array}{l} A_{Pi}=1+b_{pi}^{2}\\ B_{Pi}=-2\left(x_{Spi}+b_{pi}y_{Spi}-a_{pi}b_{pi}\right)\\ C_{Pi}=x_{Spi}^{2}+y_{Spi}^{2}-2a_{pi}y_{Spi}+a_{pi}^{2}-l_{c}^{2}+\left[{\left(x_{i}-x_{Spi}\right)}^{2}+{\left(y_{i}-y_{Spi}\right)}^{2}\right]\\ x_{Spi}=\frac{b_{pi}}{1+b_{pi}^{2}}\left(y_{i}+\frac{1}{b_{pi}}x_{i}-a_{pi}\right)\\ y_{Spi}=\frac{1}{1+b_{pi}^{2}}\left(b_{pi}^{2}y_{i}+b_{pi}x_{i}+a_{pi}\right) \end{array}$$

The *x_Spi_* and *y_Spi_* values are the coordinates of the *S_pi_* point (Figure 1), which lies at the intersection of line (8) with the line perpendicular to it passing through point *i*.

The “+” sign in Formulas (9) and (10) occurs when the values of *Y* abscissas of the measured route points are increasing, while the “−” sign is valid for decreasing abscissas. When operating with the local coordinate system, this note applies to abscissa *x*. 

For a chord drawn from point *i* backward, the interval in which the end of the chord occurs is determined by the points *q_i_* and *q_i_* + 1 (Figure 1). We determine it in the same way as in the case of the forward chord, successively checking the distances between point *i* and consecutive measurement points, going in decreasing numerical order. These distances are
(11)$$l_{(i-k)÷i}=\sqrt{{\left(x_{i}-x_{i-k}\right)}^{2}+{\left(y_{i}-y_{i-k}\right)}^{2}},\;\;k=1,2,\ldots$$

After each step of the calculations, we check whether the condition $l_{(i-k)÷i}\geq l_{c}$ has been met. The first value of *i* − *k* that meets the condition is marked as *q_i._* Since the coordinates of points *q_i_* and *q_i_* + 1 are known, it is possible to write the equation of a straight line that goes through these points analytically. This equation is as follows:(12)$$y=a_{qi}+b_{qi}x$$

As can be seen from Figure 1, the end of the back chord (i.e., point *Q_i_*) lies on the straight line described by Equation (12) at a distance *l_c_* from point *i*. Thus, it is the point of intersection of the circle with radius *l_c_* and center at point *i* with straight line (12). The coordinates of the *Q_i_* point are determined from the following formulas:(13)$$x_{Qi}=\frac{-B_{Qi}\pm \sqrt{B_{Qi}^{2}-4A_{Qi}C_{Qi}}}{2A_{Qi}}$$
(14)$$y_{Qi}=a_{qi}+b_{qi}\frac{-B_{Qi}\pm \sqrt{B_{Qi}^{2}-4A_{Qi}C_{Qi}}}{2A_{Qi}}$$
where
$$\begin{array}{l} A_{Qi}=1+b_{qi}^{2}\\ B_{Qi}=-2\left(x_{Sqi}+b_{qi}y_{Sqi}-a_{qi}b_{qi}\right)\\ C_{Qi}=x_{Sqi}^{2}+y_{Sqi}^{2}-2a_{qi}y_{Sqi}+a_{qi}^{2}-l_{c}^{2}+\left[{\left(x_{i}-x_{Sqi}\right)}^{2}+{\left(y_{i}-y_{Sqi}\right)}^{2}\right]\\ x_{Sqi}=\frac{b_{qi}}{1+b_{qi}^{2}}\left(y_{i}+\frac{1}{b_{qi}}x_{i}-a_{qi}\right)\\ y_{Sqi}=\frac{1}{1+b_{qi}^{2}}\left(b_{qi}^{2}y_{i}+b_{qi}x_{i}+a_{qi}\right) \end{array}$$

The *x_Sqi_* and *y_Sqi_* values are the coordinates of the *S_qi_* point (Figure 1), which lies at the intersection of line (12) with the line perpendicular to it passing through point *i*.

The “−” sign in Formulas (13) and (14) occurs when the abscissa values of the measured route points are increasing, while the “+” sign is valid for decreasing abscissas. Thus, what we are dealing with here is the opposite situation to the case of a forward chord.

Having the Cartesian coordinates of point *i* (obtained from measurements) and the coordinates of the ends of virtual chords drawn forward and backward, we are able to determine the curvature values at a given measurement point. A forward chord connects point *i* with point *P_i_* and its coordinates are given by Formulas (9) and (10). It is described by the equation
(15)$$y=y_{i}-\frac{y_{Pi}-y_{i}}{x_{Pi}-x_{i}}x_{i}+\frac{y_{Pi}-y_{i}}{x_{Pi}-x_{i}}x$$

The angle of inclination of straight line (15) is
(16)$$\Theta_{i÷Pi}=\Theta_{i}^{\left(+\right)}=\arctan\frac{y_{Pi}-y_{i}}{x_{Pi}-x_{i}}$$

The backward chord connects point *i* with point *Q_i_*, and its coordinates are given by Formulas (13) and (14). It is described by the equation
(17)$$y=y_{Qi}-\frac{y_{i}-y_{Qi}}{x_{i}-x_{Qi}}y_{Qi}+\frac{y_{i}-y_{Qi}}{x_{i}-x_{Qi}}x$$

The angle of inclination of straight line (17) is
(18)$$\Theta_{Qi÷i}=\Theta_{i}^{\left(-\right)}=\arctan\frac{y_{i}-y_{Qi}}{x_{i}-x_{Qi}}$$

In this situation, the curvature value at a given measurement point is determined using the formula
(19)$$\kappa_{i}=\frac{\Theta_{i}^{\left(+\right)}-\Theta_{i}^{\left(-\right)}}{l_{c}}$$

A positive value of the curvature determined by Formula (19) corresponds to a curve with convexity directed downward, and a negative value a curve with convexity directed upward.

The presented procedure is sequential and consists in using the given calculation formulas. Determination of the curvature value does not require the development of special computer programs, and the entire operation can be carried out, for example, in a spreadsheet.

## 3. Determination of the Curvature Values for the Test Section

The procedure for identifying the geometric layout of the exploited railway route has been illustrated with a calculated example consisting of a test section with a length of 5.5 km. Cartesian coordinates of individual measurement points were determined at intervals of about 5 m, and the maximum error of this operation was ±10 mm. Figure 2 shows the visualization of the course of the route on the test section in the local coordinate system.

As can be seen, the test section consists of six straight lines *S_j_* and five arcs *A_j_*_÷*j*+1_ connecting these lines with each other. The course of the route is quite gentle, and the existing geometric layout in the horizontal plane allows for the relatively fast speed of trains. Since, as it turns out, the values of the radius of the circular arcs are greater than 1400 m here, there is a possibility of using a speed of *V* = 160 km/h. 

In paper [26] it was shown that the curvature graphs of the axis of an exploited railway track clearly differ from the graphs obtained for model layouts; they have a less regular, oscillatory character, which results from measurement error and the deformations of the ballasted track [32,33,34]. However, this did not prevent the basic geometrical parameters of the measured layout from being estimated. 

When choosing the length of the mobile chord that would correspond to the situation on the test section, the recommendations formulated in [28] were followed. Based on the analysis carried out there, it was clearly demonstrated that the length of the chord used to determine the curvature in an exploited railway track should depend on the value of the radius of the circular arc. The criterion for choosing the length of the chord was the minimization of the deviations of the curvature value from the theoretical course, i.e., zero on straight sections of the track, horizontal (but not zero) along the lengths of circular arcs, and changing linearly on transition curves. The following approximate lengths of *l_c_* have been proposed, depending on the range of *R_CA_* values:for *R_CA_* ≤ 600 m     *l_c_* = 20 m;for 600 < *R_CA_* ≤ 1000 m  *l_c_* = 30 m;for 1000 < *R_CA_* ≤ 1400 m   *l_c_* = 40 m;for *R_CA_* > 1400 m      *l_c_* = 50 m.

In the considered case, a virtual chord with a length of *l_c_* = 50 m was used to determine the horizontal curvature. The calculations carried out used the procedure described in Section 2. Figure 3 shows the obtained curvature diagram along the length of the test section, and Table 1 shows a fragment of the calculations carried out in the area of curve *A*_1÷2_.

The curvature diagram in Figure 3 identifies the geometric layout along almost its entire length. Minor disturbances do not affect the overall assessment of the situation. On straight segments, the curvature is equal to zero, and on circular arcs it has a fixed value (resulting from the value of the radius). Variable curvature occurs only on transition curves and, as can be seen, it is linear there. The linear *L* coordinate allows one to specify the location of individual geometric elements.

In this situation, full identification of the layout still requires determination of boundary points between straight sections, transition curves, and circular arcs. These are the so-called segmentation points. The straight line *S_j_* is connected to the beginning of the transition curve *TC_j_*_÷*j*+1_(*a*), and on the other side of curve *A_j_*_÷*j*+1_ the straight line *S_j_*_+1_ is connected to the beginning of transition curve *TC_j_*_÷*j*+1_(*b*). In turn, the ends of the transition curves *TC_j_*_÷*j*+1_(*a*) and *TC_j_*_÷*j*+1_(*b*) determine the location of the beginning and end of the circular arc *CA_j_*_÷*j*+1_, respectively. 

Therefore, the moving chord method does not allow for direct determination of the segmentation points of the geometric layouts—it becomes necessary to carry out an additional procedure. It becomes clear, however, that the transition curves play the main role here, and determining the location of their extreme points makes it possible to determine the segmentation points and lengths of individual curves. The next section of this paper focuses on these problems, examining the individual arc sections separately.

## 4. Determining the Location of Segmentation Points

### 4.1. Arc Section A_1÷2_

Figure 4 shows the curvature diagram along the length of arc section *A*_1÷2_. The average value of curvature on a circular arc is also marked. The values *κ_i_* for i∈〈147;167〉 (corresponding to L∈〈730;830〉 m) were used for determining $\bar{\kappa_{CA}}$. The result was $\bar{\kappa_{CA}}$ = 0.0005561 rad/m, with standard deviation of *σ_CA_* = 0.000009372 rad/m (which is 1.684% of the mean value). The calculations show—as the inverse $\bar{\kappa_{CA}}$—the radius of circular arc *CA*_1÷*2*_ equal to 1798.233 m.

There are transition curves on both sides of the circular arc. They can be easily identified on the *κ*(*L*) diagram: the curvature ordinates oscillate around a linear course. In order to determine the linear coordinates of the beginnings and ends of the transition curves, it is necessary to determine the coefficients of the least squares lines describing the regions of the *κ*(*L*) graph with variable curvature values. Least squares lines in the form
*κ*(*L*) = *a* + *b L*(20)
determine the linear coordinates of their points of intersection with curvature diagrams on straight sections of the track (coordinates *L_BTC_* of the beginnings of curves) and on the circular arc sections (coordinates *L_ETC_* of ends of curves).

For the beginning of a given transition curve (*BTC* point), the value of curvature *κ* = 0, hence its linear coordinate is
(21)$$L_{BTC}=-\frac{a}{b}$$
and for the end of the curve (*ETC* point), the curvature value *κ =*
$\bar{\kappa_{CA}}$, so its linear coordinate is equal to
(22)$$L_{ETC}=\frac{\bar{\kappa_{CA}}-a}{b}$$

The values of the determined *L_BTC_* and *L_ETC_* coordinates directly result in the length of the transition curve.
(23)$$l_{TC}=\left|L_{ETC}-L_{BTC}\right|$$

The further calculation procedure will take place in the *x*, *y* rectangular coordinate system, therefore appropriate Cartesian coordinates of the segmentation points should also be determined. For the linear coordinate *L_BTC_* one should find such a range of measurement points i∈〈i,i+1〉 that $L_{BTC}\in \langle L_{i},L_{i+1}\rangle$. The abscissa *x_BTC_* and ordinate *y_BTC_* can now be determined from the following formulas:(24)$$x_{BTC}=x_{i}+\frac{x_{i+1}-x_{i}}{L_{i+1}-L_{i}}\left(L_{BTC}-L_{i}\right)$$
(25)$$y_{BTC}=y_{i}+\frac{y_{i+1}-y_{i}}{L_{i+1}-L_{i}}\left(L_{BTC}-L_{i}\right)$$

In an analogous way, the values of the abscissa *x_ETC_* of the transition curve end, as well as the corresponding *y_ETC_* ordinate, are determined. 

Figure 5 shows the effects of identifying the transition curve *TC*_1÷2_(*a*) located on the left side of the geometrical layout under consideration. 

In the conducted analysis, the values of *κ_i_* were used for i∈〈121;140〉 (which corresponds to L∈〈600;695〉 m). The equation of the curvature was obtained by the method of least squares.
*κ*(*L*) = −0.00273287 *+* 0.0000046745 *L*

On the basis of Formulas (21) and (22), the linear coordinates of *L_BTC_* and *L_ETC_* were determined. They are: *L_BTC_* = 584.630 m and *L_ETC_* = 703.595 m. Hence, based on Equation (23), the length of the considered transition curve is 118.964 m.

To determine the Cartesian coordinates *x_BTC_* and *y_BTC_*, it must be taken into account that $L_{BTC}\in \langle 580;585\rangle$ m, so the limits of the given interval are designated by *i* = 117 and *i* = 118. Since in this case *x_i_* = 57.423 m, *y_i_* = 26.007 m, and *x_i_*_+1_ = 584.406 m, *y_i_*_+1_ = 26.229 m, Formulas (24) and (25) show that *x_BTC_* = 584.038 m and *y_BTC_* = 26.212 m.

To determine *x_ETC_* and *y_ETC_* coordinates, it is necessary to take into account that $L_{ETC}\in \langle 700;705\rangle$ m, so the limits of a given interval are determined by *i* = 141 and *i* = 142. Since *x_i_* = 699.226 m, *y_i_* = 32.533 m, and *x_i_*_+1_ = 704.222 m, *y_i_*_+1_ = 32.920 m, based on Formulas (24) and (25) we see that *x_ETC_* = 702.818 m and *y_ETC_* = 32.811 m.

Figure 6 shows the identification of transition curve *TC*_1÷2_(*b*) located on the right side of the geometrical layout under consideration.

Using the *κ_i_* values for i∈〈175;197〉 (corresponding to L∈〈870;980〉 m), the following curvature equation was obtained:*κ*(*L*) = 0.00435742 − 0.0000044463 *L*

Using the Formulas (21) and (22), the linear coordinates of *L_BTC_* and *L_ETC_* were determined, which are: *L_BTC_* = 980.001 m and *L_ETC_* = 854.931 m. This results in the length of the given transition curve *l_TC_* = 125.070 m.

Since $L_{BTC}\in \langle 980;985\rangle$ m, the limits of the interval are determined by *i* = 197 and *i* = 198. In this case, *x_i_* = 975.927 m, *y_i_* = 73.976 m, and *x_i_*_+1_ = 980.837 m, *y_i_*_+1_ = 74.946 m, hence, based on Formulas (24) and (25), *x_BTC_* = 975.928 m and *y_BTC_* = 73.976 m. In turn, $L_{ETC}\in \langle 850;855\rangle$ m, so the limits of an interval are determined by *i* = 171 and *i* = 172. Since *x_i_* = 848.162 m, *y_i_* = 50.116 m, and *x_i_*_+1_ = 853.095 m, *y_i_*_+1_ = 50.906 m, from Formulas (24) and (25) it is obtained that *x_ETC_* = 853.030 m and *y_ETC_* = 50.895 m.

### 4.2. Arc Section A_2÷3_

Figure 7 shows the curvature diagram along the length of arc section *A*_2÷3_, and Figure 8 and Figure 9 show the identification of transition curves *TC*_2÷3_(*a*) and *TC*_2÷3_(*b*). 

Assuming an analogous procedure as in Section 4.1, the radius of the circular arc *CA*_2÷3_ was determined to be 1639.433 m. The linear coordinates of the transition curve *TC*_2÷3_(*a*) are: *L_BTC_* = 1546.687 m and *L_ETC_* = 1673.502 m. Hence, based on Equation (23), the length of this transition curve *l_TC_* = 127.215 m. For transition curve *TC*_2÷3_(*b*) the following values were obtained: *L_BTC_* = 1949.973 m and *L_ETC_* = 1823.556 m, so the length of this curve is 126.417 m. The corresponding values of Cartesian coordinates *x_BTC_* and *y_ETC_* for both curves are included in the list in Table 2. 

### 4.3. Arc Section A_3÷4_

Figure 10 shows the curvature diagram along the length of arc section *A*_3÷4_, and Figure 11 and Figure 12 show the identification of transition curves *TC*_3÷4_(*a*) and *TC*_3÷4_(*b*). 

Assuming an analogous procedure as in Section 4.1, the radius of the circular arc *CA*_3÷4_ was determined to be 1460.686 m. The linear coordinates of the transition curve *TC*_3÷4_(*a*) are: *L_BTC_* = 2539.613 m and *L_ETC_* = 2681.172 m. Hence, based on Equation (23), the length of this transition curve *l_TC_* = 141.559 m. For transition curve *TC*_3÷4_(*b*) the following values were obtained: *L_BTC_* = 3071.951 m and *L_ETC_* = 2932.738 m; therefore the length of this curve is 139.213 m. The corresponding values of Cartesian coordinates *x_BTC_* and *y_ETC_* for both curves are included in the list in Table 2. 

### 4.4. Arc Section A_4÷5_

Figure 13 shows the curvature diagram along the length of arc section *A*_3÷4_, and Figure 14 and Figure 15 show the identification of transition curves *TC*_4÷5_(*a*) and *TC*_4÷5_(*b*). 

Assuming an analogous procedure as in Section 4.1, the radius of the circular arc *CA*_4÷5_ was determined to be 1546.006 m. The linear coordinates of the transition curve *TC*_4÷5_(*a*) are: *L_BTC_* = 3734.707 m and *L_ETC_* = 3869.805 m. Hence, based on Equation (23), the length of this transition curve *l_TC_* = 135.080 m. For transition curve *TC*_2÷3_(*b*) the following values were obtained: *L_BTC_* = 4209.072 m and *L_ETC_* = 4065.732 m; therefore, the length of this curve is 143.340 m. The corresponding values of Cartesian coordinates *x_BTC_* and *y_ETC_* for both curves are included in the list in Table 2. 

### 4.5. Arc Section A_5÷6_

Figure 16 shows a graph of the curvature along the length of arc segment *A*_5÷6_. This graph significantly differs from the previously studied cases. That is because of the short length of the circular arc in relation to the moving chord used. If the arc length is less than *l_c_*, the κ(*L*) diagram lacks the curvature values from which the arithmetic mean (and, consequently, the radius of the circular arc) can be determined. 

In the studied situation, the radius of the circular arc was determined by the method of measuring the horizontal arrows. The assumed value of *R* = 1920 m was used to determine the segmentation points, assuming linearity of the curvature course on both transition curves. The discussed situation is shown in Figure 17.

In the case of the *TC*_5÷6_(*a*) transition curve (left in Figure 17), the values of *κ_i_* for i∈〈948;963〉 (which corresponds to L∈〈4735;4810〉 m) were used. The equation of curvature was obtained by the least squares method.
*κ*(*L*) = 0.02315629 − 0.0000049039 *L*

Based on Formulas (21) and (22), the linear coordinates of *L_BTC_* and *L_ETC_* were determined. They are as follows: *L_BTC_* = 4722.052 m and *L_ETC_* = 4828.261 m. Therefore, based on Equation (23), the length of the considered transition curve is 106.209 m. 

In the analysis of the *TC*_5÷6_(*b*) transition curve (on the right in Figure 17), the values of *κ_i_* for i∈〈975;993〉 (corresponding to L∈〈4870;4960〉 m) were taken into account. The equation of curvature was obtained by the least squares method.
*κ*(*L*) = −0.02401088 + 0.0000048312 *L*

The determined linear coordinates of *L_BTC_* and *L_ETC_* are: *L_BTC_* = 4969.917 m and *L_ETC_* = 4862.111 m. Hence, the length of the studied transition curve is *l_TC_* = 107.805 m. It also becomes possible to determine the length of the circular arc as the difference between the ends of both transition curves: it is 33.851 m.

Cartesian coordinates *x_BTC_* and *y_ETC_* of the transition curves shown in Figure 17 were determined in the same way as the transition curve *TC*_1÷2_. Their values are included in the collective list in Table 2. 

## 5. Full Identification of the Geometrical Layout

The coordinates of the segmentation points determined in Section 4 allow one to fully identify the geometrical layout present on the test section. It has been presented in the form of a collective list (Table 2).

The presented calculation procedure allows one to fully use the measured coordinates of the track axis. Apart from visualizing the course of the route and obtaining a general orientation of the existing geometric elements (as in Figure 2), these coordinates allow one to directly determine the horizontal curvature (Figure 3). It is possible thanks to the use of a new method for determining the curvature. The moving chord method allowed for comprehensive identification of the existing geometric elements (straight lines, circular arcs, and transition curves), along with the determination of all segmentation points.

## 6. Conclusions

Determining the geometrical shape of a railway route is based on the measured coordinates of the track axis in an appropriate reference system. The currently used measurement methods allow for high precision and a significant reduction in time consumption. These methods are constantly developing—apart from classic geodetic techniques, satellite measurements are used, both in stationary and mobile versions. Determining the coordinates of the track axis enables the visualization of a given railway route, giving a general orientation of its location. However, since the purpose of the measurements is to determine the geometrical parameters (i.e., identification) of the measured route, some additional actions should be taken; in the standard approach, this will mean measuring horizontal arrows.

Meanwhile, the measured coordinates of the track axis allow one to directly determine the existing horizontal curvature, without the need for additional measurements. It is possible thanks to the development of a new method for determining the curvature, the so-called method of the moving chord. It allows one to comprehensively identify the existing geometric elements (straight sections, circular arcs, and transition curves) along with the determination of segmentation points that define the connections of these elements with each other.

This paper presents a detailed algorithm for determining the curvature of the track axis with the use of the moving chord method, and then discusses the procedure for identifying the geometric layout of an exploited railway route on the basis of the determined curvature. The conducted activities have been illustrated with calculation examples, covering a 5.5 km long test section with five areas of directional change. It showed a sequential procedure that led to the solution of the given problem. Based on the curvature diagram, the coordinates of the segmentation points, which allow one to fully identify the geometrical layout present in the test section, were determined. 

The moving chord method does not allow for directly determining the segmentation points of the geometric layout; it is necessary to carry out an additional procedure. It is easy to notice, however, that the main role here is played by the transition curves, and the determination of the location of their extreme points enables one to determine segmentation points and lengths of individual curves. The significant difficulty of identifying transition curves makes attempting to use the directional angle of the route to determine the parameters of the geometric layout [35] not fully effective. It is not possible to ensure sufficient accuracy of calculations with the necessary numerical differentiation of the directional angle in the areas where the transition curve connects with a straight section and with a circular arc. This has been explained in [27].

## Figures and Tables

**Figure 1 sensors-23-00274-f001:**
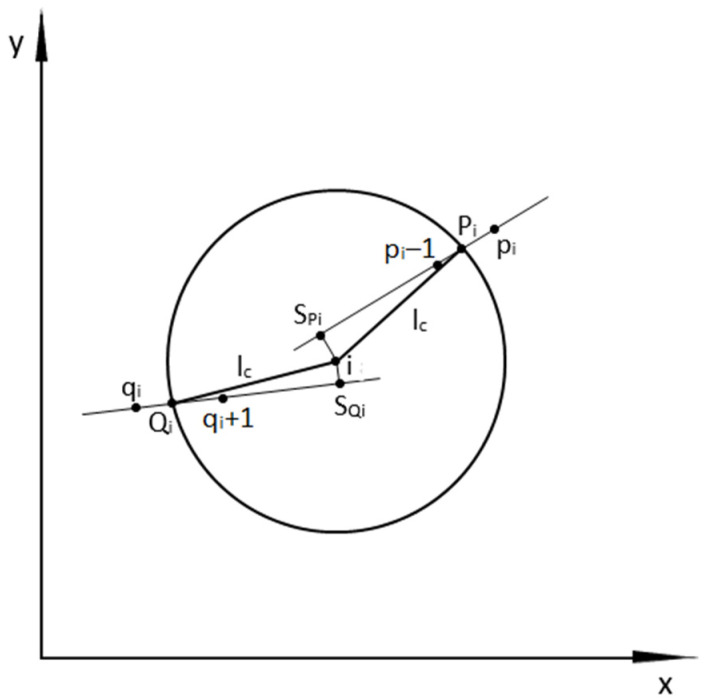
Explanation of the method for determining the curvature value at any measurement point using the moving chord method.

**Figure 2 sensors-23-00274-f002:**
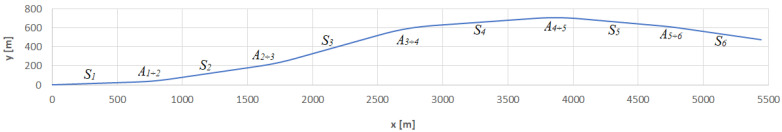
The course of the route on the test section in the local coordinate system.

**Figure 3 sensors-23-00274-f003:**
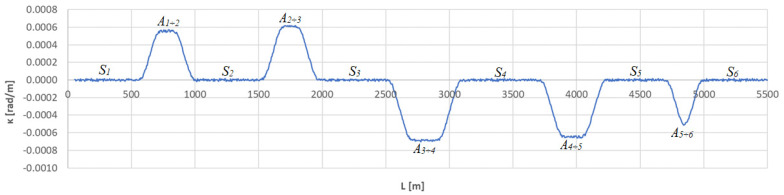
Curvature diagram along the length of the test section obtained using a chord of length *l_c_* = 50 m.

**Figure 4 sensors-23-00274-f004:**
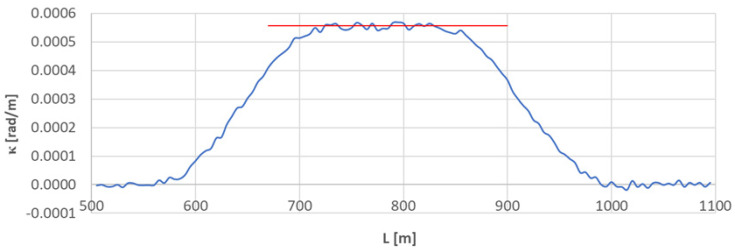
Curvature diagram along the length of arc segment *A*_1÷2_ obtained using a chord of length *l_c_* = 50 m (the value of the average curvature on the circular arc is marked in red).

**Figure 5 sensors-23-00274-f005:**
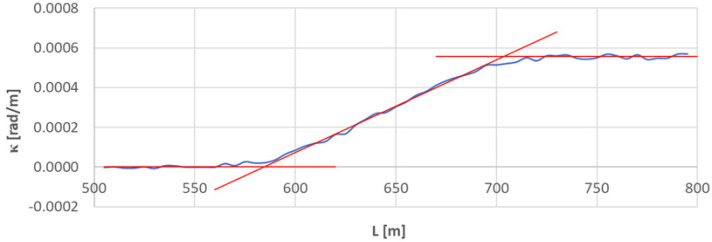
Identification of transition curve *TC*_1÷2_(*a*) on the test section (curvature diagram obtained using a chord of length *l_c_* = 50 m).

**Figure 6 sensors-23-00274-f006:**
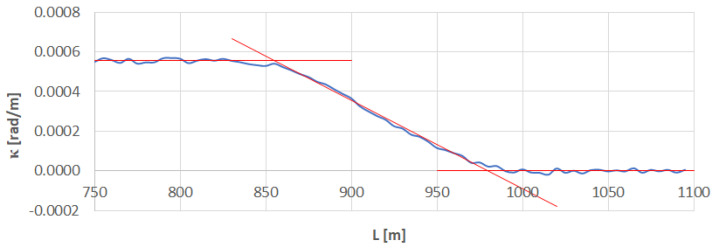
Identification of transition curve *TC*_1÷2_(*b*) on the test section (curvature diagram obtained using a chord of length *l_c_* = 50 m).

**Figure 7 sensors-23-00274-f007:**
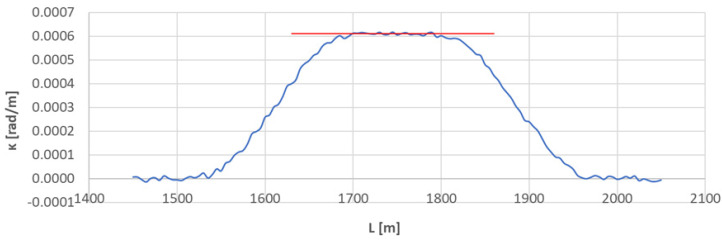
Curvature diagram along the length of arc segment *A*_2÷3_ obtained using a chord of length *l_c_* = 50 m (the value of the average curvature on the circular arc is marked in red).

**Figure 8 sensors-23-00274-f008:**
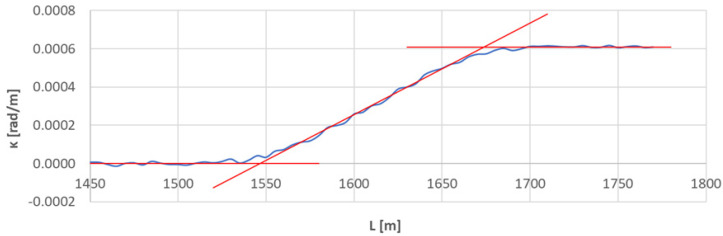
Identification of transition curve *TC*_2÷3_(*a*) on the test section (curvature diagram obtained using a chord of length *l_c_* = 50 m).

**Figure 9 sensors-23-00274-f009:**
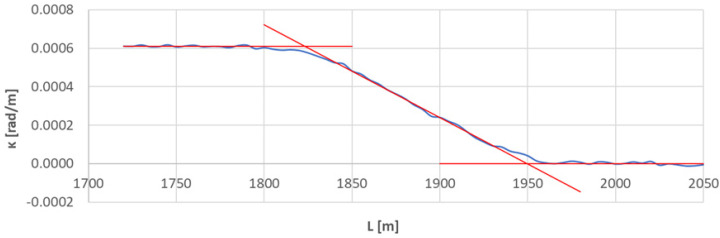
Identification of transition curve *TC*_2÷3_(*b*) on the test section (curvature diagram obtained using a chord of length *l_c_* = 50 m).

**Figure 10 sensors-23-00274-f010:**
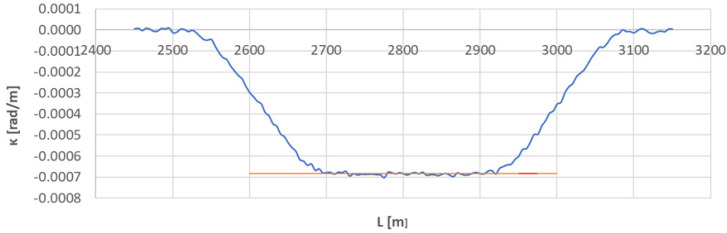
Curvature diagram along the length of arc segment *A*_3÷4_ obtained using a chord of length *l_c_* = 50 m (the value of the average curvature on the circular arc is marked in red).

**Figure 11 sensors-23-00274-f011:**
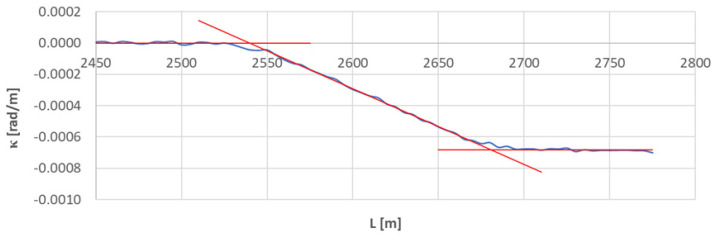
Identification of transition curve *TC*_3÷4_(*a*) on the test section (curvature diagram obtained using a chord of length *l_c_* = 50 m).

**Figure 12 sensors-23-00274-f012:**
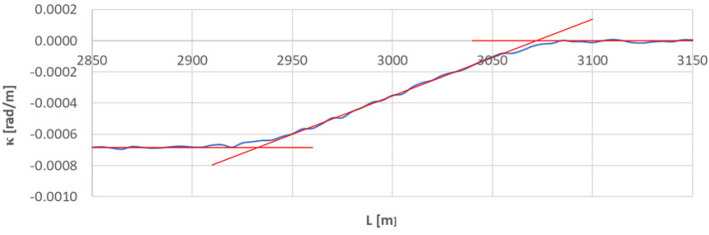
Identification of transition curve *TC*_3÷4_(*b*) on the test section (curvature diagram obtained using a chord of length *l_c_* = 50 m).

**Figure 13 sensors-23-00274-f013:**
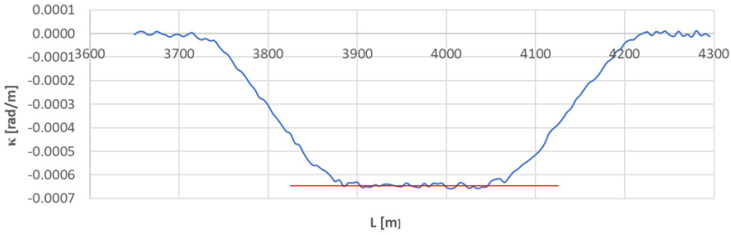
Curvature diagram along the length of arc segment *A*_4÷5_ obtained using a chord of length *l_c_* = 50 m (the value of the average curvature on the circular arc is marked in red).

**Figure 14 sensors-23-00274-f014:**
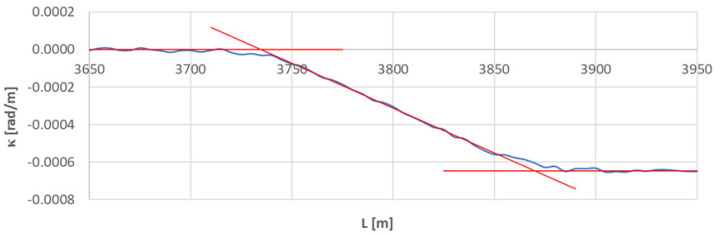
Identification of transition curve *TC*_4÷5_(*a*) on the test section (curvature diagram obtained using a chord of length *l_c_* = 50 m).

**Figure 15 sensors-23-00274-f015:**
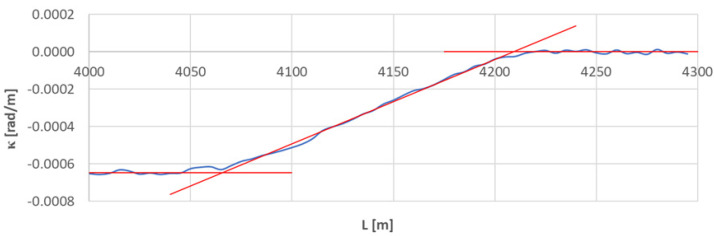
Identification of transition curve *TC*_4÷5_(*b*) on the test section (curvature diagram obtained using a chord of length *l_c_* = 50 m).

**Figure 16 sensors-23-00274-f016:**
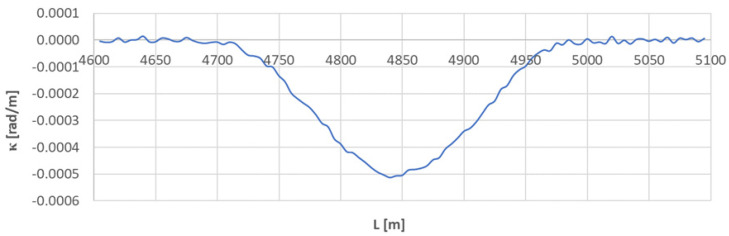
Curvature diagram along the length of arc segment *A*_5÷6_ obtained using a chord of length *l_c_* = 50 m.

**Figure 17 sensors-23-00274-f017:**
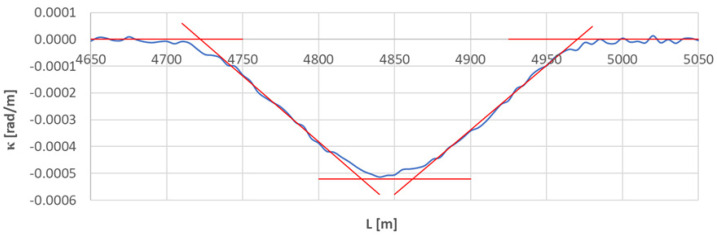
Identification of the geometric layout consisting of transition curves *TC*_5÷6_(*a*) and *TC*_5÷6_(*b*).

**Table 1 sensors-23-00274-t001:** Chosen fragment of the performed calculations of curvature *κ_i_* in the area of arch *A*_1÷2_.

Point *i*	*L_i_* [m]	*x_i_* [m]	*y_i_* [m]	*x_Pi_* [m]	*y_Pi_* [m]	Θ*_i÷Pi_* [rad]	*x_Qi_* [m]	*y_Qi_* [m]	Θ*_Qi÷i_* [rad]	*κ_i_* [rad/m]
153	760	758.967	38.063	808.588	44.201	0.12309	709,193	33.313	0.09513	0.00055911
154	765	763.940	38.629	813.546	44.890	0.12557	714.182	33.720	0.09834	0.00054459
155	770	768.916	39.179	818.502	45.602	0.12881	719.169	34.160	0.10055	0.00056514
156	775	773.872	39.776	823.443	46.312	0.13109	724.143	34.583	0.10404	0.00054093
157	780	778.834	40.368	828.585	47.055	0.13414	729.119	35.040	0.10676	0.00054746
158	785	783.812	40.975	833.344	47.802	0.13696	734.112	35.507	0.10959	0.00054756
159	790	788.774	41.581	838.285	48.60	0.14004	739.086	36.011	0.11164	0.00056806
160	795	793.731	42.214	843.221	49.33	0.14288	744.058	36.505	0.11442	0.00056911
161	800	798.679	42.863	848.150	50.14	0.14553	749.022	37.015	0.11724	0.00056584

**Table 2 sensors-23-00274-t002:** List of segmentation points on the test section.

Segm.	Point Coordinates			Characteristics of Segmentation Point
Point	*L* [m]	*x* [m]	*y* [m]	
1	0.000	0.000	0.000	Beginning of the test section (beginning of the straight line *S*_1_; *l* = 584.630 m)
2	584.630	584.038	26.212	End of straight line *S*1 Beginning of transition curve *TC*_1÷2_(*a*); *l* = 118.964 m
3	703.595	702.818	32.811	End of transition curve *TC*_1÷2_(a) Beginning of circular arc *CA*_1÷2_; *R* = 1798.233 m
4	854.931	853.030	50.895	End of circular arc *CA_1÷2_*End of transition curve *TC*_1÷2_(*b*); *l* = 125.070 m
5	980.001	975.928	73.976	Beginning of transition curve *TC*_1÷2_(*b*) Beginning of straight line *S*_2_; *l* = 566.686 m
6	1546.687	1531.726	184.552	End of straight line *S*_2_ Beginning of transition curve *TC*_2÷3_(*a*); *l* = 127.215 m
7	1673.902	1656.174	210.943	End of transition curve *TC*_2÷3_(*a*) Beginning of circular arc *CA*_2÷3_; *R* = 1639.433 m
8	1823.556	1799.220	252.387	End of circular arc *CA*_2÷3_ End of transition curve *TC*_2÷3_(*b*); *l* = 126.417 m
9	1949.973	1918.572	295.991	Beginning of transition curve *TC*_2÷3_(*b*) Beginning of straight line *S*_3_; *l* = 589.640 m
10	2539.613	2469.412	506.330	End of straight line *S*_3_ Beginning of transition curve *TC*_3÷4_(*a*); *l* = 141.559 m
11	2681.172	2602.432	554.722	End of transition curve *TC*_3÷4_(*a*) Beginning of circular arc *CA*_3÷4_; *R* = 1460.686 m
12	2932.738	2847.048	612.063	End of circular arc *CA*_3÷4_ End of transition curve *TC*_3÷4_(*b*); *l* = 139.213 m
13	3071.951	2985.378	627.615	Beginning of transition curve *TC*_3÷4_(*b*) Beginning of straight line *S*_4_; *l* = 662.756 m
14	3734.707	3645.076	691.310	End of straight line *S*_4_ Beginning of transition curve *TC*_4÷5_(*a*); *l* = 135.098 m
15	3869.805	3779.705	702.380	End of transition curve *TC*_4÷5_(*a*) Beginning of circular arc *CA*_4÷5_; *R* = 1546.006 m
16	4065.732	3975.482	700.341	End of circular arc *CA*_4÷5_ End of transition curve *TC*_4÷5_(*b*); *l* = 143.340 m
17	4209.072	4118.013	685.267	Beginning of transition curve *TC*_4÷5_(*b*) Beginning of straight line *S*_5_; *l* = 512.980 m
18	4722.052	4627.295	623.819	End of straight line *S*_5_ Beginning of transition curve *TC*_5÷6_(*a*); *l* = 106.209 m
19	4828.261	4732.624	610.136	End of transition curve *TC*_5÷6_(*a*) Beginning of circular arc *CA*_5÷6_; *R* = 1920.000 m
20	4862.111	4766.053	604.850	End of circular arc *CA*_5÷6_ End of transition curve *TC*_5÷6_(*b*); *l* = 107.805 m
21	4969.917	4872.026	585.089	Beginning of transition curve *TC*_5÷6_(*b*) Beginning of straight line *S*_6_; *l* = 580.083 m
22	5550.000	5441.295	473.501	End of test section (the end of straight line *P*_6_)

## Data Availability

Cartesian coordinate values of the considered geometric layout of the track that support the findings of this study are available from the author upon reasonable request.

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
