# Peer review of "The Procedure of Identifying the Geometrical Layout of an Exploited Railway Route Based on the Determined Curvature of the Track Axis"

_sensors, 2022, doi:10.3390/s23010274_

Round 1

Reviewer 1 Report

The manuscript “The Procedure of Identifying the Geometrical Layout of an Exploited Railway Route Based on the Determined Curvature of the Track Axis” deals with an problem of identifying the geometric layout of an exploited railway route from its Cartesian coordinates obtained by direct measurements. A comprehensive algorithm is presented that takes into account the track longitudinal coordinate system and the corresponding double differentiation by the help of moving chord method. The obtained curvature along the track corresponds to the track measurement results by machine and manual measurements. The geometrical layout is identified by fitting of the lines corresponding to circular and transition curves.

The manuscript topic fits to the scope of the Journal. It is well structured. The conclusions correspond to the research done. The paper could be an important contribution in the field track measurements.

However, there is a number of unclear points that need additional explanation and improvement according to the comments:

11.     The general aim of the element identification is not fully clear. The element position on the track is usually known before. And the curvature cannot be used for the maintenance due to high error of the absolute measurements.

22.    The chord 50 m causes that the transition zones are longer than the real ones and the short elements can be smoothed (Figure 16.). On the other side, a short chord would cause high variation of the curvature. What could be a criterion of the optimal chord length?

33.    The manuscript review includes high part of the own studies compared to the overall number of the references. It is recommended to improve the literature review with new relevant sources that are also presented in MDPI journals (Identification of sleeper support conditions…, Mechanism of sleeper dynamic impact in zones with unsupported sleepers…Optical rail surface crack detection…).

Reviewer 2 Report

The paper presents a algorithm for determining the curvature of the track axis with the use of a moving chord method, and then discusses the procedure for identifying the geometric layout of an exploited railway route on the basis of the determined curvature.

The study is interesting, it can be published after minor revise.

(1)    Formula derivations appear in the introduction, that’s not appropriate.

(2)    All the symbols appearing in the formula should be explained.

(3)    determination of the curvature value of the test section shall be described in more detail.

(4)    How to further verify the effectiveness of the proposed algorithm?
